# TAKE ONE GRAM OF NEURAL FEATURES, GET ENHANCED GROUP ROBUSTNESS

## ABSTRACT

Predictive performance of machine learning models trained with empirical risk minimization (ERM) can degrade considerably under distribution shifts. In particular, the presence of spurious correlations in training datasets leads ERM-trained models to display high loss when evaluated on minority groups not presenting such correlations in test sets. Extensive attempts have been made to develop methods improving worst-group robustness. However, they require group information for each training input or at least, a validation set with group labels to tune their hyperparameters, which may be expensive to get or unknown a priori. In this paper, we address the challenge of improving group robustness *without group annotations during training*. To this end, we propose to partition automatically the training dataset into groups based on Gram matrices of features extracted from an identification model and to apply robust optimization based on these pseudo-groups. In the realistic context where no group labels are available, our experiments show that our approach not only improves group robustness over ERM but also outperforms all recent baselines.

## 1 INTRODUCTION

Empirical Risk Minimization (ERM) is the most standard machine learning formulation, which assumes that training and testing samples are independent and identically distributed (Vapnik, 1991). While academic datasets are mainly built to respect this assumption, practical settings display more challenging configurations with distribution shifts. Among different types of shifts, training data can be affected by selection biases and confounding factors, also called *spurious correlations* (Woodward, 2005; Duchi et al., 2019)

Imagine crowd-sourcing an image dataset of camels and cows (Beery et al., 2018). Due to selection biases, a large majority of cows stand in front of grass environment and camels in the desert. A simple way to differentiate cows from camels would be to classify the background, an undesirable shortcut that ERM will naturally exploit. Consequently, ERM may perform poorly on minority groups that do not display such spurious correlations (Hashimoto et al., 2018; Tatman, 2017; Duchi et al., 2019), e.g., a cow standing in the desert. To overcome this issue, recent works (Creager et al., 2021; Bao & Barzilay, 2022; Sohoni et al., 2020; Liu et al., 2021; Ahmed et al., 2021; Kirichenko et al., 2022) rely on two-stage schemes: first, automatic environment discovery (e.g., based on deep feature clustering); then, robust optimization based on environment pseudo-labels. *Environment* here refers to a recurring setting, not intrinsic to the object of interest, that may affect its classification, such as background, object color or object pose. However, all these approaches require the availability of ground-truth environment labels on a validation set to properly tune their hyperparameters.

This paper addresses the problem of learning a robust classifier, which, for instance, would not confuse a cow standing in the desert with a camel although not given any annotation about grass or desert. In computer vision, many identified spurious correlations are closely related to visual aspects, such as background (Beery et al., 2018), texture (Geirhos et al., 2019), image style (Hendrycks et al., 2021), physic attributes (Liu et al., 2015) or camera characteristics (Koh et al., 2021). In this work, we assume that relevant environment labels can be inferred from visual feature statistics, and demonstrate they lead to meaningful environments and robust classifiers for standard datasets used to evaluate robust classification. We propose a two-stage approach, GRAMCLUST, which first assigns a group label, i.e., a class-environment pair label, by partitioning a training dataset into clusters of

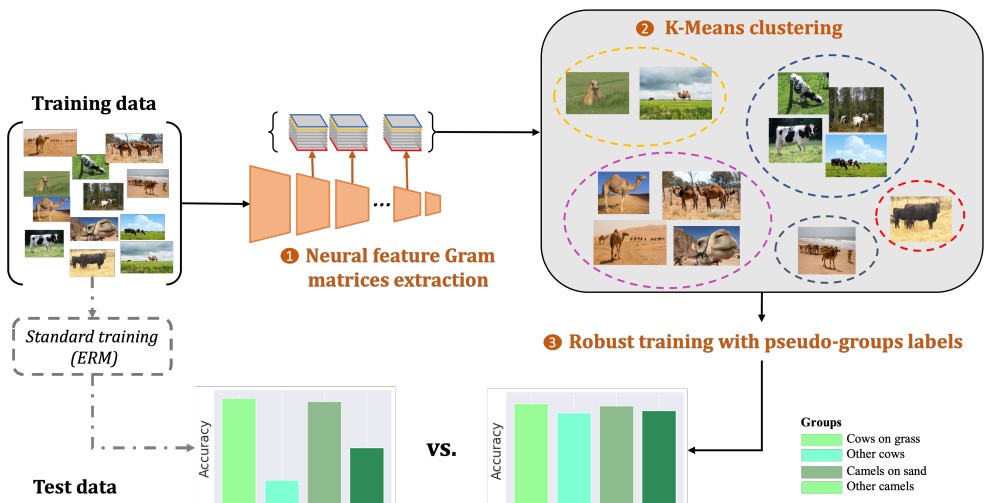

Figure 1: **Overview of the proposed GRAMCLUST approach for robust classification with unsupervised group discovery.** (1) We first extract deep image features using an identification model and (2) we cluster the training dataset based on Gram matrices of images features; (3) Then, we train the targeted classifier with a robust optimization that exploits the assigned pseudo-group labels. Consequently, GRAMCLUST properly classifies samples in minority groups, e.g. cows and camels in unusual environments – in contrast to standard Empirical Risk Minimization (ERM) training.

images with similar visual statistics and then trains a robust classifier based on these pseudo-group labels. Our approach is summarized in Fig. 1. We use Gram matrices as visual descriptive statistics, which are second-order moments of neural activation. Gram matrices are well known for displaying impressive results in style transfer techniques (Gatys et al., 2016), but more importantly for the interpretation of our approach, Li et al. (2017) demonstrate that matching Gram matrices between two groups of images is equivalent to aligning the respective distribution of each group, minimizing the Maximum Mean Discrepancy. Therefore, our method can be interpreted as grouping images into clusters of similar feature distributions that are sensible candidates for environments.

Our main contributions are as follows: (1) We introduce an easy-to-scale method to split training images among distinct pseudo-environments, based on feature Gram matrices extracted by a specifically-trained identification model; (2) GRAMCLUST alleviates the need of ground-truth group labels altogether, even in the validation set, as hyperparameters are set based on validation performance computed from our pseudo-groups; (3) Extensive experiments on various image classification datasets with spurious correlations show that GRAMCLUST outperforms all recent baselines addressing robustness without group annotation. In particular, on the realistic large-scale CelebA dataset (Liu et al., 2015), we improve worst-group test accuracy by +24.3 points.

## 2 RELATED WORK

Robustness to distribution shift (Rusak et al., 2020; Hendrycks* et al., 2020; Gulrajani & Lopez-Paz, 2021; Geirhos et al., 2019) has recently been an increasingly popular topic among machine learning researchers. Koh et al. (2021) distinguish two types of distribution shifts: *domain generalization*, where test samples come from a different distribution than training datasets, and *subpopulation shift*, where train and test distributions overlap but their relative proportion differs. With subpopulation shifts, the goal is to perform well even on the minority group, also referred to as *group robustness*. In this study, we focus on the latter form of distribution shift.

**Group robustness with group annotations.** Recent approaches propose to leverage group annotations during training to improve group robustness. IRM (Arjovsky et al., 2020) augments the standard ERM term with invariance penalties across data from different groups. Ahmed et al. (2021) promote, through a simple penalty, identical prediction behaviour across groups. Other works (Sagawa

et al., 2020a; Zhang et al., 2021) minimize explicitly the worst-group loss during training. Finally, Sagawa et al. (2020b) re-balance majority and minority groups via re-weighting and sub-sampling.

**Group robustness without group annotations.** The focus here is a more realistic setting in which group annotations are not available on the training data. Creager et al. (2021) derive a group inference objective from a trained identification model that maximizes variability across environments, and is differentiable w.r.t a distribution over group assignments. Liu et al. (2021) introduce a simple method in which environments are defined by images on which a trained identification model performs poorly. Sohoni et al. (2020) propose an unsupervised clustering algorithm in the feature space of a trained identification model. But these methods require implicitly or explicitly a small validation set with ground-truth group annotation: Liu et al. (2021) explicitly tunes its hyperparameters on a small validation set with group annotation while Creager et al. (2021) and Sohoni et al. (2020) use best hyperparameters used in the GroupDRO paper (Sagawa et al., 2020a) to minimize the worst-group accuracy on their inferred groups. These best hyperparameters were actually found using a validation set with true-group labels in the original study.

**Gram matrices.** The original work of Gatys et al. (2016) demonstrated impressive results to generate images with the style of an existing image. The style of a first image is transferred to a second one by matching Gram matrices of features extracted by a convolutional neural network. Sastry & Oore (2020) also used Gram matrices in out-of-distribution detection to identify an anomaly by comparing their values to the respective range observed over the training data. Interestingly, Li et al. (2017) demonstrate a formal equivalence between matching Gram matrices of neural activations with an $L_2$ norm and the MMD with the second-order polynomial kernel. This shows that Gram matrices are also implicitly used in the process of distribution alignment between images. This finding motivates our approach, which consists in discovering pseudo-groups using Gram matrices.

## 3    GRAMCLUST: A CLUSTERING APPROACH FOR ROBUST OPTIMIZATION

Our method, GRAMCLUST, consists of two main steps. First, we discover pseudo-environments among the images of a given dataset (see Section 3.2). Second, we train a robust classifier that leverages the inferred pseudo-environment labels to reduce classification errors due to spurious environment correlations (see Section 3.3). To discover environments, we train during a few iterations an exogenous "identification model". Then, using this model, we compute for each image its Gram matrix representation from different layers and apply random projections to reduce dimension. The resulting concatenated features are then fed to an unsupervised clustering algorithm ($k$-means) to produce pseudo-environment labels. This allows us to define pseudo-groups as the intersection of pseudo-environments and classes. Last, we train the target classifier by minimizing the standard cross-entropy classification loss on the worst pseudo-group with GroupDRO (Sagawa et al., 2020a).

### 3.1    PROBLEM FORMULATION AND NOTATIONS

Let us consider a dataset $\mathcal{D} = \{(\boldsymbol{x}_i, y_i)\}_{i=1}^N \in \mathcal{X} \times \mathcal{Y}$ of $N$ samples where $\mathcal{X}$ is the input space and $\mathcal{Y} = [\![1, K]\!]$ a set of labels. We assume the data is sampled from random variables $(\boldsymbol{X}_e, \boldsymbol{Y}_e)$ in $\mathcal{X} \times \mathcal{Y}$ with probability law $\mathbb{P}_{(\boldsymbol{X}_e, \boldsymbol{Y}_e)}$ for all $e \in [\![1, E]\!]$, where $E$ is the number of *environments*. The full dataset can then be seen as the union of subsets associated to each random variable, i.e., $\mathcal{D} = \bigcup_{e=1}^E \mathcal{D}_e$ where each $\mathcal{D}_e$ is composed of *i.i.d.* realisations of a random variable with joint probability law $\mathbb{P}_{(\boldsymbol{X}_e, \boldsymbol{Y}_e)}$. For notation purposes, we actually choose the following equivalent formulation for the dataset $\mathcal{D} = \{(\boldsymbol{x}_i, y_i, e_i)\}_{i=1}^N \in \mathcal{X} \times \mathcal{Y} \times [\![1, E]\!]$ where $e_i$ refers to the environment from which $\boldsymbol{x}_i$ and $y_i$ were sampled.

Our goal is to find a model $m$ in a given hypothesis space $\mathcal{M}$ that minimizes the error on the worst group. A *group* is defined as a set of samples both from the same class and in the same environment. Formally, we introduce group distributions:

$$\mathbb{P}_{\boldsymbol{G}_{1,1}} = \mathbb{P}(\boldsymbol{X}_1 | \boldsymbol{Y}_1 = 1), \ \cdots, \ \mathbb{P}_{\boldsymbol{G}_{E,K}} = \mathbb{P}(\boldsymbol{X}_E | \boldsymbol{Y}_E = K). \tag{1}$$

The purpose is then to solve the following objective minimization problem:

$$\underset{m \in \mathcal{M}}{\arg\min} \left\{ \max_{g \in [\![1,E]\!] \times [\![1,K]\!]} \mathbb{E}_{(\boldsymbol{x}, y) \sim \mathbb{P}_{\boldsymbol{G}_g}} \big[ \ell(m(\boldsymbol{x}), y) \big] \right\}, \tag{2}$$

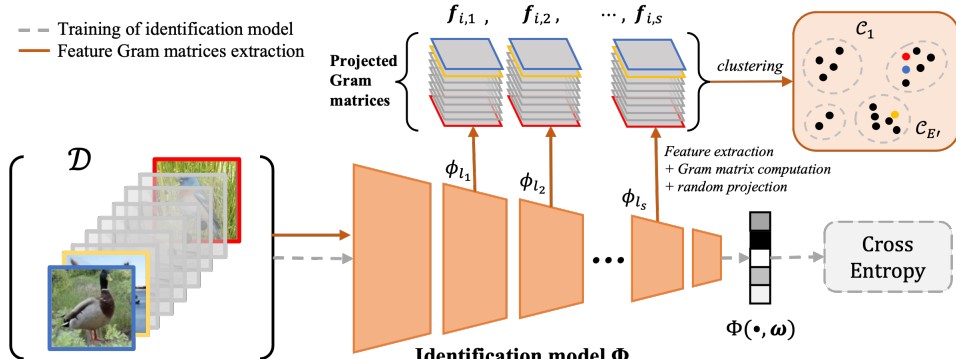

Figure 2: **Illustration of the proposed dataset partition**. An identification model $\Phi$ with parameters $\boldsymbol{\omega}$ is trained for a limited number $T$ of epochs with ERM to fit groups with easy-to-learn spurious correlations. Then, for each image $\boldsymbol{x}_i \in \mathcal{X}$, we extract intermediate features $\phi_l$ at layer $l$ and compute their Gram matrix $\mathsf{G}_l$ with a random projection. These projected Gram matrix representations are used as features to cluster the training dataset $\mathcal{D}_{\text{train}}$ in $E'$ environments.

where $\ell : \mathcal{Y} \times \mathcal{Y} \to \mathbb{R}^+$ is the cross-entropy loss between the model's prediction and the true label. Note that we have no access to any environment labels. To circumvent this issue, we first discover pseudo-environment labels, then estimate the pseudo-group distributions to be used in Eq. (2).

## 3.2  DATASET PARTITION

In this section, we describe the first stage of GRAMCLUST, which aims at environment discovery. The method is illustrated in Fig. 2.

**Identification model.** Our approach starts by initializing a convolutional neural network $\Phi$ for the classification task at hand; it is composed of $L$ layers with parameters $\boldsymbol{\omega}$ and is pre-trained on ImageNet (Deng et al., 2009). Liu et al. (2021) observed that ERM tends to fit models on data presenting easy-to-learn spurious correlations at the beginning of the learning process. It is only after a significant number of epochs that the model starts to learn more difficult patterns. Hence, we only train $\Phi$ during a few iterations, minimizing w.r.t. $\boldsymbol{\omega}$ the following empirical loss function:

$$\frac{1}{N} \sum_{i=1}^{N} \ell(\Phi(\boldsymbol{x}_i, \boldsymbol{\omega}), y_i), \tag{3}$$

where $\ell : \mathcal{Y} \times \mathcal{Y} \to \mathbb{R}^+$ is the cross-entropy loss between the model's predicted label $\Phi(\boldsymbol{x}_i, \boldsymbol{\omega})$ and the true label $y$ associated with sample $\boldsymbol{x}$.

In the following, we call $\Phi$ the *identification model* as our clustering is based on features extracted from this model. The idea is to leverage the biases learned by $\Phi$ to identify relevant environments and partition the training dataset into groups of images presenting spurious correlations, on the one hand, and groups of images free from these correlations on the other hand. Hence, after this initial training and in the rest of the paper, the parameters $\boldsymbol{\omega}$ of the identification model $\Phi$ are frozen.

**Features Gram matrices.** We denote the feature map of an image $\boldsymbol{x}$ at layer $l$ of $\Phi$ by $\phi_l(\boldsymbol{x}) \in \mathbb{R}^{M_l \times C_l}$, where $C_l$ is the number of channels and $M_l$ is the spatial size of the feature map. For each image $\boldsymbol{x}_i \in \mathcal{X}$, we extract its feature maps at $S \leqslant L$ different and fixed layers $\{l_1, \cdots l_S\}$, and compute the Gram matrices defined as:

$$\mathsf{G}_l(\boldsymbol{x}_i) = \frac{1}{M_l} \, \phi_l(\boldsymbol{x}_i)^\mathsf{T} \phi_l(\boldsymbol{x}_i) \in \mathbb{R}^{C_l \times C_l}, \quad l \in \{l_1, \cdots l_S\}. \tag{4}$$

Given input $\boldsymbol{x}_i$ and identification model $\Phi$, the Gram matrix of its feature $\phi_l(\boldsymbol{x}_i)$ encodes visual correlations via an inner product between each pair of vectorized feature maps. In visual style transfer (Gatys et al., 2016), these Gram matrices have been shown to encode the "style" of an image, that is, loosely speaking, its textures and color palette, by contrast with its "structure".

**Clustering with $k$-means.** For each image $\boldsymbol{x}_i$, we vectorize and normalize its $S$ associated Gram matrices: $\boldsymbol{f}_{i,l} = \text{vec}(\mathsf{G}_l(\boldsymbol{x}_i))/\left\|\text{vec}(\mathsf{G}_l(\boldsymbol{x}_i))\right\|_2 \in \mathbb{R}^{C_l^2}$. The normalization permits us to balance the contributions of the different Gram matrices in the clustering loss. Each image $\boldsymbol{x}_i$ is thus encoded by the vector $\boldsymbol{f}_i = [\boldsymbol{f}_{i,1}, \ldots, \boldsymbol{f}_{i,S}] \in \mathbb{R}^C$, where $C = \sum_l C_l^2$. Relying on the assumption that environments can be inferred from visual feature statistics, we propose to discover $E'$ environments, $E'$ being a parameter as $E$ is unknown, by clustering the $N$ training images into $E'$ clusters $\{\mathcal{C}_1, \ldots \mathcal{C}_{E'}\}$ via $k$-means clustering, i.e., by computing a solution to:

$$\min_{\{\mathcal{C}_1, \ldots \mathcal{C}_{E'}\}} \sum_{e=1}^{E'} \frac{1}{2\left|\mathcal{C}_e\right|} \sum_{\left\{i,j \mid \mathbf{x}_i, \mathbf{x}_j \in \mathcal{C}_e \times \mathcal{C}_e\right\}} \left\|\boldsymbol{f}_i - \boldsymbol{f}_j\right\|_2^2, \tag{5}$$

where $\left\|\boldsymbol{f}_i - \boldsymbol{f}_j\right\|_2^2 = \sum_{l=1}^S \left\|\boldsymbol{f}_{i,l} - \boldsymbol{f}_{j,l}\right\|_2^2$.

**Scaling with random projections.** Storing all these vectors and computing distances between them in a high-dimensional space is computationally and memory expensive on large datasets. We overcome this difficulty by projecting the vectors $\boldsymbol{f}_{i,l}$ in a lower-dimensional space as proposed by Achlioptas (2003). Given a size $\ell_0$, we build a matrix $\mathsf{P} \in \mathbb{R}^{\ell_0 \times C}$ whose entries $\mathsf{P}_{mn}$ are the realisation of independent random variables: $\mathsf{P}_{mn} = 1$ or $\mathsf{P}_{mn} = -1$ with probability $1/2$. Then we compute

$$\tilde{\boldsymbol{f}}_{i,l} = \frac{1}{\sqrt{\ell_0}}\mathsf{P}\boldsymbol{f}_{i,l} \tag{6}$$

and substitute $\tilde{\boldsymbol{f}}_{i,l}$ for $\boldsymbol{f}_{i,l}$ in Eq. (5). We justify this choice by the fact that this projection preserves the distances $\left\|\boldsymbol{f}_{i,l} - \boldsymbol{f}_{j,l}\right\|_2^2$ involved in the $k$-means objective of Eq. (5). Indeed, let $\epsilon \in ]0,1[$ and $\ell_0 \propto \log(N)$, then with high probability,[1]

$$(1-\epsilon)\left\|\boldsymbol{f}_{i,l} - \boldsymbol{f}_{j,l}\right\|_2 \leqslant \left\|\tilde{\boldsymbol{f}}_{i,l} - \tilde{\boldsymbol{f}}_{j,l}\right\|_2 \leqslant (1+\epsilon)\left\|\boldsymbol{f}_{i,l} - \boldsymbol{f}_{j,l}\right\|_2, \tag{7}$$

for all $(i,j) \in [\![1,N]\!]^2$. In practice, we choose $\ell_0 = \lfloor 100\log(N) \rfloor$ which yields dimensions for $\tilde{\boldsymbol{f}}_{i,l}$ much lower than typical values of $C_l$. We remark that this choice of projection is independent of all $\boldsymbol{f}_{i,l}$ and thus can be defined and fixed before any feature extraction.

## 3.3 ROBUST OPTIMIZATION WITH PSEUDO-GROUP LABELS

Given these estimated environments, we define their intersection with classes as "pseudo-groups". Formally, given the predicted environment $\hat{e}_i \in [\![1,E']\!]$, of image $i$, its pseud-group label is $\hat{g}_i = (\hat{e}_i, y_i) \in [\![1,E']\!] \times [\![1,K]\!]$.

Going back to Eq. (2), the distributions over the groups $\mathbb{P}_{\boldsymbol{G}_{\hat{g}}}$ are estimated by

$$\hat{\mathbb{P}}_{\boldsymbol{G}_{\hat{g}}} = \delta(\mathcal{G}(\hat{g})) \qquad \text{for all } \hat{g} \in [\![1,E']\!] \times [\![1,K]\!], \tag{8}$$

where $\delta$ is the Dirac distribution, and

$$\mathcal{G}(1,1) = \{(\boldsymbol{x}_i, y_i), i \in [\![1,N]\!] \mid y_i = 1, x_i \in \mathcal{C}_1\} \tag{9}$$

$$\cdots$$

$$\mathcal{G}(E',K) = \{(\boldsymbol{x}_i, y_i), i \in [\![1,N]\!] \mid y_i = K, x_i \in \mathcal{C}_{E'}\} \tag{10}$$

are the sets of images and labels associated with the pseudo-group labels.

Each training point $\boldsymbol{x}_i \in \mathcal{X}$ is now associated with a class label $y_i$ and a pseudo-group annotation $\hat{g}_i$. We train a robust classifier $h$ with parameters $\boldsymbol{\theta}$ by minimizing the worst-group risk on the training dataset (Sagawa et al., 2020a):

$$\hat{\boldsymbol{\theta}} \in \arg\min_{\boldsymbol{\theta}} \left\{ \max_{\hat{g} \in [\![1,E']\!] \times [\![1,K]\!]} \frac{1}{|\mathcal{G}(\hat{g})|} \sum_{(\boldsymbol{x},y) \in \mathcal{G}(\hat{g})} \left[\ell(h(\boldsymbol{x}, \boldsymbol{\theta}), y)\right] \right\}, \tag{11}$$

where the loss $\ell : \mathcal{Y} \times \mathcal{Y} \to \mathbb{R}^+$ remains the cross-entropy between the predicted label $h(\boldsymbol{x}, \boldsymbol{\theta})$ of the robust classifier and the true label $y$ associated with sample $\boldsymbol{x}$.

---

[1]We let the reader refer to Theorem 1.1 in (Achlioptas, 2003) for the exact expression of this probability as a function of $\epsilon$, $N$ and $\ell_0$.

### 3.4 Model selection via cross-validation on validation data

Setting relevant hyperparameters is important in optimization algorithms to ensure a proper convergence. Hyperparameters tuning is performed with cross-validation using a held-out subset of training data. With robust optimization, worst-group accuracy of the final classifier is the go-to metric for model selection. Previous approaches rely on *true* group labels of the validation set to define and assess performance on the worst group. In contrast, we do not rely on such a prior information. We partition the validation set using the clusters found on the training set and we conduct cross-validation based on the resulting pseudo-groups. In our experiments, we observe that this type of model selection is effective to achieve proper group robustness.

## 4 Experiments

In this section, we evaluate the capacity of GRAMCLUST to improve group robustness on image classification datasets with spurious correlations. In Section 4.2, we empirically show that it outperforms other baselines addressing robustness without group annotation on three datasets. We then present in Section 4.3 an empirical analysis of our approach, including the importance of using Gram matrices as visual features, the impact of the choice of layers to extract features from, and the impact of the number of clusters. The code is available with the supplementary material.

### 4.1 Setup

**Datasets**. We experiment with three image classification datasets on which previous works evaluate worst-group performance. **Waterbirds** (Sagawa et al., 2020a) is a dataset composed of images combining bird photographs from the CUB dataset (Welinder et al., 2010) with background scenes taken from the Places365 dataset (Zhou et al., 2018). The target labels ("landbirds"/"waterbirds") are spuriously correlated with the background images ("land"/"water"). The train set is composed of 4,795 images and the validation and test set are respectively composed of 1,199 and 2,897 images. **CelebA** (Liu et al., 2015) is a celebrity face dataset with 202,599 images. Sagawa et al. (2020a) considered the task of classifying the hair color of the individual as "blond" or "not blond". The authors observed that there exists a spurious correlation between the hair color and the gender ("male" or "female") of a person. In fact, in the dataset, only 2% of blond people are male. We use the official train-val-test split from Liu et al. (2015). **COCO-on-Places-224** is a dataset of 10 segmented MS COCO (Lin et al., 2014) objects superimposed on scenes from the Places365 dataset. The train set has 7,200 training images — 800 images per category — and validation and test sets are composed of 900 images — 100 images per category. Unlike Waterbirds, multiple backgrounds have spurious correlations with the object classification (Ahmed et al., 2021). We rebuilt this dataset based on the code provided by Ahmed et al. (2021)[2] but with images resized to $224 \times 224$ (instead of $64 \times 64$ in the original paper, which considerably degrades visual features of objects and background).

**Baselines**. We compare our approach against the standard **ERM** baseline and recent methods that aim at robust predictions across groups without the use of train group annotations: **EIIL** (Creager et al., 2021), **GEORGE** (Sohoni et al., 2020), and **JTT** (Liu et al., 2021). We also include robust methods that use train group annotations: **IRM** (Arjovsky et al., 2020), importance weighting and **GroupDRO** (Sagawa et al., 2020a). The latter methods and ERM were already implemented and we took care to reproduce results for all methods. Our results with baselines are in line with those reported respectively in each original paper. Note that our approach and GroupDRO share the same robust optimization objective (Eq. (2)). Hence, GRAMCLUST would boil down to GroupDRO if discovered pseudo-groups were to match exactly the ones annotated in the dataset.

**Training details**. All methods, including ours, use a ResNet-50 (He et al., 2016) architecture pretrained on ImageNet as the *robust classifier*. Models are optimized using SGD with momentum. For GroupDRO and ERM, we use the hyperparameters reported by Sagawa et al. (2020a) on Waterbirds and CelebA datasets. Further training details are available in Appendix A. Note that hyperparameters have been selected with the use of a validation set with group labels. Regarding our approach, we select a VGG-19 (Simonyan & Zisserman, 2015) architecture for the *identification model* $\Phi$. Although dating from 2015, VGG-19 is still the go-to architecture for applications involving Gram

---

[2]https://github.com/Faruk-Ahmed/predictive_group_invariance

Table 1: **Comparative results on Waterbirds, CelebA and COCO-on-Places-224.** Worst-group (*w-g*) and average (*avg*) test accuracies (% mean and std.) for Waterbirds and CelebA datasets; systematically-shifted (*shift*) and in-distribution (*in-dis*) test-set accuracies (% mean and std.) for COCO-on-Places-224 dataset. Experiments are with ResNet-50 models. Underlined and **bold** type indicate respectively best and per-block best performance (with significance $p < 0.05$ according to paired t-test on five runs). Methods are grouped according to their need for ground-truth group labels on train and/or val set; proposed GRAMCLUST-*cv* is the only one requiring none.

| Method | Group labels train | val | Waterbirds (w-g) | (avg) | CelebA (w-g) | (avg) | COCO-on-P (shift) | (in-dis) |
|---|---|---|---|---|---|---|---|---|
| ERM | | ✓ | $65.0_{\pm 2.7}$ | $\underline{97.3}_{\pm 0.1}$ | $42.4_{\pm 1.5}$ | $\underline{94.8}_{\pm 0.1}$ | $71.9_{\pm 0.3}$ | $\underline{95.5}_{\pm 0.1}$ |
| IRM (Arjovsky et al., 2020) | ✓ | ✓ | $77.4_{\pm 0.3}$ | $\underline{97.3}_{\pm 0.1}$ | $75.1_{\pm 0.6}$ | $94.5_{\pm 0.1}$ | $78.8_{\pm 0.3}$ | $\mathbf{95.1}_{\pm 0.2}$ |
| Importance weighting | ✓ | ✓ | $74.4_{\pm 0.6}$ | $\underline{97.4}_{\pm 0.1}$ | $72.4_{\pm 1.4}$ | $94.4_{\pm 0.2}$ | $71.7_{\pm 0.5}$ | $93.7_{\pm 0.2}$ |
| GroupDRO (Sagawa et al., 2020a) | ✓ | ✓ | $\mathbf{83.9}_{\pm 0.3}$ | $96.8_{\pm 0.1}$ | $\mathbf{85.7}_{\pm 2.0}$ | $93.7_{\pm 0.2}$ | $\underline{\mathbf{79.0}}_{\pm 0.4}$ | $95.2_{\pm 0.2}$ |
| EIIL (Creager et al., 2021) | | ✓ | $78.7_{\pm 0.3}$ | $96.9_{\pm 0.1}$ | - | - | $68.5_{\pm 0.4}$ | $94.8_{\pm 0.3}$ |
| GEORGE (Sohoni et al., 2020) | | ✓ | $76.2_{\pm 2.0}$ | $95.7_{\pm 0.5}$ | $53.7_{\pm 1.3}$ | $\mathbf{94.6}_{\pm 0.2}$ | $71.6_{\pm 0.3}$ | $95.1_{\pm 0.1}$ |
| JTT[3] (Liu et al., 2021) | | ✓ | $82.9_{\pm 0.3}$ | $96.4_{\pm 0.2}$ | $56.0_{\pm 0.7}$ | $93.6_{\pm 0.0}$ | $69.2_{\pm 0.4}$ | $94.7_{\pm 0.3}$ |
| GRAMCLUST-*orig* (Ours) | | ✓ | $\underline{\mathbf{85.3}}_{\pm 1.1}$ | $96.6_{\pm 0.1}$ | $\mathbf{77.9}_{\pm 2.2}$ | $94.2_{\pm 0.2}$ | $\mathbf{72.4}_{\pm 0.4}$ | $95.0_{\pm 0.2}$ |
| GRAMCLUST-*cv* (Ours) | | | $\underline{\mathbf{85.3}}_{\pm 1.1}$ | $96.6_{\pm 0.1}$ | $\mathbf{80.3}_{\pm 1.9}$ | $93.4_{\pm 0.1}$ | $\mathbf{73.2}_{\pm 0.3}$ | $\underline{\mathbf{95.3}}_{\pm 0.3}$ |

matrices such as image style transfer (Zhang et al., 2022; Höllein et al., 2022; Xie et al., 2022). Results of our approach include two types of model selection via cross-validation: (i) based on a validation set with true-group annotations (GRAMCLUST-*orig*), and (ii) based on pseudo-group labels (GRAMCLUST-*cv*) predicted by our clustering (see Section 3.4).

**Metrics**. We report worst-group and average test accuracy for Waterbirds and CelebA datasets. On COCO-on-Places-224, we follow the evaluation protocol proposed by Ahmed et al. (2021) and report predictive performance on the in-distribution test set, which follows the same distribution as the training set, and on the systematically-shifted test set, where the spurious correlations have been removed and COCO objects are composed with uniformly-sampled random backgrounds.

## 4.2 COMPARATIVE RESULTS

We report quantitative comparisons on Waterbirds, CelebA and COCO-on-Places-224 in Table 1. We observe that GRAMCLUST improves worst-group test accuracy over ERM baseline on Waterbirds and CelebA and systematic generalisation on COCO-on-Places224. More importantly, GRAMCLUST-*cv* achieves state-of-the-art performance on group robustness compared to all methods that do not use group labels on the training set. This results show empirically that our proposed approach, using Gram matrices of feature to discover pseudo-groups, which are then used for robust optimization and hyperparameter cross-validation, is very effective for group robustness. It also supports that Gram matrices are well suited to capture various types of dataset biases (background for Waterbirds, physical attribute in CelebA, multiple backgrounds in COCO-on-Places-224). For instance, on Waterbirds, GRAMCLUST-*cv* achieves $85.3\%$ worst-group accuracy compared to the second-best method, JTT, which reaches $82.9\%$. The gap is even more pronounced on CelebA where our approach outperforms JTT by 24.3 pts. CelebA constitutes an interesting dataset to evaluate the scalability of methods as the training dataset is composed of 200k images. For instance, we were not able to scale EIIL on this dataset. Note that GRAMCLUST-*orig* uses the same hyperparameters as EIIL, GEORGE and JTT for robust training of the target classifier from predicted group labels, and still displays significant improvements on the three datasets in terms of worst-group accuracy. Liu et al. (2021) reported results that were obtained with early stopping thanks to a small validation set annotated with group labels. The authors selected models before convergence (around epoch 3) with low average accuracy on the test set but high worst-group accuracy. We argue that it is not a suitable property for a model and prefer models with high accuracy both in average and on the worst group of the test set. Surprisingly, GRAMCLUST-*cv* and GRAMCLUST-*orig* outperform GroupDRO on Waterbirds with $85.3\%$ vs. $83.9\%$, while the latter method uses true-group labels during training. Our intuition is that it may be due to the ambiguity of the background in some Waterbirds

---

[3]Results with JTT differ from the original paper as the scores that we report correspond to models trained without early-stopping.

Table 2: **Study of the clustering features**. Results in worst-group (Waterbirds, CelebA) and systematically-shifted (COCO-on-P) test-set accuracies (%). Gram matrices show to be the most effective type of information to obtain improved group robustness.

| Visual features | Architecture | Layer | Waterbirds | CelebA | COCO-on-P |
|---|---|---|---|---|---|
| Standard | ResNet-50 | *AvgPool* | $76.2_{\pm 2.0}$ | $53.7_{\pm 1.3}$ | $71.6_{\pm 0.3}$ |
| MeanVar | VGG-19 | *Conv5_1* | $\textbf{85.3}_{\pm 1.2}$ | $69.8_{\pm 1.0}$ | $71.4_{\pm 0.5}$ |
| Gram matrix | VGG-19 | *Conv5_1* | $\textbf{85.3}_{\pm 1.1}$ | $\textbf{77.9}_{\pm 2.2}$ | $\textbf{72.4}_{\pm 0.4}$ |

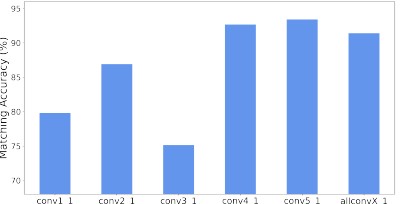

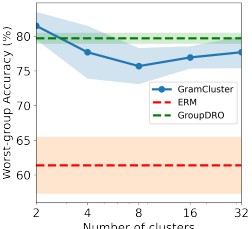

Figure 3: **Impact of the layer choice to extract features**. Results in matching accuracy on the validation set for GRAMCLUST on Waterbirds.

Figure 4: **Impact of the number of clusters.** Results in worst-group val accuracies of GRAMCLUST on Waterbirds.

images. We further discuss this result in Section 4.5. Overall, these results show that our pseudo-groups on the validation set are relevant to select good hyperparameters and more importantly, that GRAMCLUST does not require any group labels during training to achieve group robustness.

### 4.3 STUDY OF THE CLUSTERING FEATURES

In this section, we compare the performance obtained when clustering images with different visual features. In neural style transfer, Huang & Belongie (2017) proposed the channel-wise mean and variance of image features, instead of Gram matrices as in (Gatys et al., 2016). We thus compare the use of such features (*'MeanVar'*) against our use of Gram matrices. We also compared our use of VGG-19 features with the direct use of the penultimate representation of a ResNet-50 identification model (*'AvgPool'*). Recall that although our features for group identification are extracted using a VGG-19, our robust classifier is a ResNet-50. One may wonder if using directly the deepest features before the classification head in a ResNet-50 (*'Standard'*) could be better than using VGG-19 features. For a fair comparison, we trained the robust classifier with the same hyperparameters for each method, which are consistent with those found by Sagawa et al. (2020a) with GroupDRO.

The results are available in Table 2 for Waterbirds, CelebA and COCO-on-Places-224. Using the penultimate layer of a ResNet-50 as visual features for the clustering produces poorer performance than Gram matrices of VGG-19 features in every configuration. *MeanVar* reaches test worst-group accuracy on-par with *Gram matrix* on Waterbirds but degrades significantly performance on CelebA: $69.8\%$ in average compared to $77.9\%$ with *Gram matrix*. Gram matrices provide more information than *MeanVar* as their diagonals already contain the information about the channel-wise mean and variance of the deep features (see Eq. (4)). This show that when scaling on large datasets such as CelebA, keeping all the correlations between different channels is important for group robustness.

### 4.4 CLUSTERING ANALYSIS

**Effect of the selected layers for features.** We evaluate the impact of the selection of VGG-19 layers to extract the features in the clustering stage. To this end, we study the matching of the predicted environments to the true environment labels on the validation set. The assignment problem is solved via Hungarian matching (Kuhn, 1955) and we measure the global matching accuracy across all validation samples, where matching accuracy is the percentage of samples whose predicted group corresponds to its true group. In Fig. 3, we compare results on Waterbirds using either one of the five layers commonly used in neural style transfer (*conv1_1*, *conv2_1*, *conv3_1*, *conv4_1*, *conv5_1*) or using all layers together. Experiments show that: (i) Features from deeper layers correlate with better matching accuracy; (ii) Our approach is robust to the choice of deep layers either taken to-

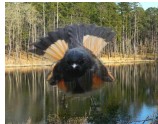 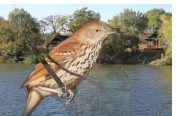 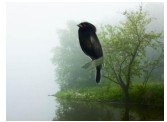 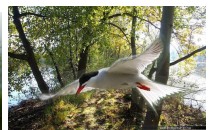 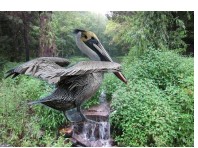 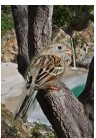

(a) True land background predicted as water background by GRAMCLUST

(b) True water background predicted as land background by GRAMCLUST

Figure 5: **Example of confusing samples in Waterbirds dataset, wrongly predicted by GRAM-CLUST.** (a) Samples of confusing land-background images predicted as water background; (b) Samples of confusing land-background images predicted as water background. In each case, the actual image background is confusing due to the joint presence of elements reflecting land background (forest, heavy vegetation, sand) and water background (water surface, rainfalls, mist).

gether (*allconvX_1*) or individually such as *conv4_1* and *conv5_1*; (iii) Using *conv5_1* outperforms selecting all traditional style layers and results in the highest score of $93.41\%$. We found consistent conclusions on the CelebA dataset. Results are reported in Appendix B.

**Impact of the number of clusters.** We study the impact of the number of clusters as hyperparameter in the clustering algorithm. Worst-group accuracy on the validation set for $E' \in \{2, 4, 8, 16, 32\}$ clusters are reported in Fig. 4 for Waterbirds datasets. Overall, our method is robust to a variation in the number of clusters: GRAMCLUST with higher numbers of clusters produces a slight drop in performance but still outperforms ERM. It also has on-par performance with GroupDRO.

## 4.5 DISCUSSION ABOUT RESULTS OF GRAMCLUST VS. GROUPDRO ON WATERBIRDS

Comparative results in Table 1 show that GRAMCLUST-*orig* outperforms GroupDRO on the Waterbirds dataset. The difference between the approaches lies in the usage of true-group labels on the training dataset for GroupDRO while GRAMCLUST-*orig* leverages its predicted pseudo-groups. This results might be surprising given that the evaluation is performed on true test group labels and that the two methods share the same robust optimization algorithm and hyperparameters. We intuit that this behavior, which occurs only on Waterbirds, is related to the group labels in the dataset. In Fig. 5, we show some examples of confusing images that were not correctly assigned with our predicted group labels with GRAMCLUST. These images are taken from the set of mismatches between true-group labels and our pseudo-group labels after the Hungarian matching. We can see that some of these samples present dominant characteristic elements from land background, such as heavy vegetation and sand, while being labeled as water background. Conversely, some samples labeled as land background display a high percentage of water surfaces in the image. As mentioned in Section 4.1, the Waterbird dataset was created by combining bird photographs with background scenes taken from the Places365 dataset. But the latter dataset is composed of very diverse images which might not reflect the expected background for a category. This unwanted behavior outlines the difficulty of manually annotating groups and raises the need for creating benchmarks including datasets with spurious correlations from non-artificial, real-world data, such as hair color/gender bias observed in CelebA. It also motivates further research on the automatic discovery of groups in data, as proposed in our method.

## 5 CONCLUSION

In this paper, we introduce GRAMCLUST, a two-stage method that first partitions a training dataset into clusters via $k$-means clustering based on Gram matrices computed from image features, which are extracted from a identification model trained to catch spurious correlations in a biased dataset. This first stage is then followed by learning a robust classifier which minimizes the error on the worst pseudo-group labels previously discovered. GRAMCLUST demonstrates to be an effective approach to tackle group robustness and outperforms every baseline on standard datasets with spurious correlations. The usage of Gram matrices of features is crucial to capture pertinent visual statistics of the image and enables a relevant partition for robust training. Our approach also alleviates the need to label a validation set of images with group information and is able to tune its hyperparameters in an unsupervised fashion by applying its clustering algorithm on the validation set.

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

# A    IMPLEMENTATION DETAILS

This section focuses on implementation details used to produce the results in the main text of our paper. The code that we used is provided along with this appendix. Our implementation builds upon the WILDS framework[4] released wby Koh et al. (2021).

## A.1    CONSTRUCTION OF COCO-ON-PLACES-224

We generated the dataset using the code[5] of Ahmed et al. (2021) but, as explained in the main paper, we modified it to produce images of size $224 \times 224$ instead of $64 \times 64$. The reader can refer to the appendix of (Ahmed et al., 2021) for more details regarding the generation of the COCO-on-Places dataset.

## A.2    DETAILS ABOUT ROBUST OPTIMIZATION

We trained all models on one NVIDIA® V100 Tensor Core with 16GB of memory, using PyTorch 1.10 and CUDA 10.2.

We used the implementations of IRM (Arjovsky et al., 2020), Importance Weighting and GroupDRO (Sagawa et al., 2020a) available in WILDS (Koh et al., 2021), our own implementations of JTT (Liu et al., 2021) and of GEORGE (Sohoni et al., 2020) (while making sure that we could reproduce the original performance on Waterbirds and CelebA), and the official implementation[6] of EIIL (Creager et al., 2021). Concerning EIIL, we recall that we were not able to make this method scale to large datasets such as CelebA.

For all methods, we used a ResNet-50 (He et al., 2016) architecture trained using stochastic gradient descent with momentum (SGD-M) and $L_2$ regularization, but without any learning rate scheduler. We used a momentum of $0.9$ and a batch size of $128$ for all datasets and all methods. The learning rate $\eta$ and $L_2$ regularization parameters $\lambda$ are set as detailed below.

JTT, GEORGE, EIIL, GRAMCLUST all use GroupDRO (Sagawa et al., 2020a) as robust optimization step. On Waterbirds and CelebA, we did not redo any grid search and used the hyperparameters found in (Sagawa et al., 2020a). These hyperparameters were optimized using a small validation set annotated with true group labels. To produce the results on COCO-on-Places-224, we performed our own grid search using the annotated validation set. We considered values of $\eta$ and $\lambda$ close to those used in (Sagawa et al., 2020a): $\lambda \in \{10^{-4}, 10^{-2}, 10^{-1}, 1\}$ and $\eta \in \{10^{-5}, 5 \cdot 10^{-5}, 10^{-4}\}$. The best hyperparameters for GroupDRO are summarized in Table 3.

To ensure fair comparisons, we also performed the same grid search over $\eta$ and $\lambda$ for ERM, IRM and Importance Weighting. The best hyperparameters for ERM and IRM are summarized for each dataset in Table 4 and Table 5, respectively. Note that they correspond to those reported in (Sagawa et al., 2020a) for Waterbirds and CelebA.

Table 3: **SGD-M hyperparameters for GroupDRO training**.

| SGD-M hyperparameters | Waterbirds | CelebA | COCO-on-Places-224 |
|---|---|---|---|
| Learning rate $\eta$ | $10^{-5}$ | $10^{-5}$ | $5 \cdot 10^{-5}$ |
| $L_2$ regularization $\lambda$ | $1.0$ | $0.1$ | $10^{-2}$ |

## A.3    GROUP DISCOVERY DETAILS

For GRAMCLUST, we follow standard practice of neural style transfer (Gatys et al., 2016) and use the VGG-19 (Simonyan & Zisserman, 2015) architecture for the identification model. This model is trained during 1 epoch on the training dataset with ERM using a batch size of 128 and SGD-M.

---

[4]https://github.com/p-lambda/wilds
[5]https://github.com/Faruk-Ahmed/predictive_group_invariance
[6]https://github.com/ecreager/eiil

Table 4: **SGD-M hyperparameters for ERM training**.

| SGD-M hyperparameters | Waterbirds | CelebA | COCO-on-Places-224 |
|---|---|---|---|
| Learning rate $\eta$ | $10^{-4}$ | $10^{-4}$ | $10^{-4}$ |
| $L_2$ regularization $\lambda$ | $10^{-3}$ | $10^{-4}$ | $10^{-4}$ |

Table 5: **SGD-M hyperparameters for IRM training**.

| SGD-M hyperparameters | Waterbirds | CelebA | COCO-on-Places-224 |
|---|---|---|---|
| Learning rate $\eta$ | $10^{-4}$ | $10^{-5}$ | $5 \cdot 10^{-5}$ |
| $L_2$ regularization $\lambda$ | $10^{-3}$ | $0.1$ | $0.1$ |

Among usual layers used to compute representations in neural style transfer, we observed improved performance by selecting deeper layers in the network (see Section 4.4). Consequently, for each dataset, we consistently extract features from the *conv5_1* layer, i.e., the first convolutional layer of block 5. Following results of Fig. 4, we set the number of clusters to 2 in our experiments.

For EIIL and GEORGE, the identification model is a ResNet-50 as used in the original methods. We train the model for 1 epoch with ERM using SGD-M, as for GRAMCLUST. Note that the activation at the output of the last layer is a sigmoid in EIIL while it is a softmax in GEORGE. As for GRAMCLUST, the best results were obtained when using 2 clusters for EIIL and GEORGE. We refer the reader to (Creager et al., 2021) and (Sohoni et al., 2020) for other implementation details specific to EIIL and GEORGE, respectively.

## A.4 CROSS VALIDATION ON PSEUDO-GROUP ANNOTATIONS

We report in Table 6 the results of our grid search on the validation set of each dataset using the *pseudo-annotations* discovered with our method, i.e., using our discovered environments instead of the ground-truth ones. Hence, the average and worst group accuracies in Table 6 are computed using the discovered pseudo-groups. The hyperparameters used in GRAMCLUST-*cv* correspond to those which yield the best worst-group accuracy in this table.

Table 6: **Grid search results on the validation sets of Waterbirds, CelebA and COCO-on-Places-224 with pseudo-group labels.** We report the worst-group ('w-g') and average ('avg') accuracies for Waterbirds and CelebA datasets, and the systematically-shifted ('shift') and in-distribution ('ind') accuracies for COCO-on-Places ('COCO-on-P') dataset.

| Method | Hyperparam. | | Waterbirds | | CelebA | | COCO-on-P | |
|---|---|---|---|---|---|---|---|---|
| | $\lambda$ | $\eta$ | w-g | avg | w-g | avg | sys | ind |
| GRAMCLUST-*cv* | 0.01 | $1 \cdot 10^{-5}$ | 74.6 | 82.4 | **86.0** | 93.2 | 62.8 | 92.3 |
| | 0.01 | $5 \cdot 10^{-5}$ | 69.2 | 79.9 | 53.5 | **94.6** | 70.7 | 76.5 |
| | 0.01 | $1 \cdot 10^{-4}$ | 70.0 | 80.6 | - | - | 78.5 | 82.7 |
| | 0.1 | $1 \cdot 10^{-5}$ | 75.4 | 82.6 | 85.6 | 93.7 | **78.7** | 83.3 |
| | 0.1 | $5 \cdot 10^{-5}$ | 73.8 | 82.4 | 85.0 | 89.1 | 70.4 | 76.4 |
| | 0.1 | $1 \cdot 10^{-4}$ | 76.9 | 85.8 | - | - | 76.2 | 81.2 |
| | 1 | $1 \cdot 10^{-5}$ | **80.8** | **86.4** | - | - | 65.5 | 72.6 |
| | 1 | $5 \cdot 10^{-5}$ | 0.0 | 23.1 | - | - | 0.1 | 11.1 |
| | 1 | $1 \cdot 10^{-4}$ | 0.0 | 23.1 | - | - | 0.2 | 11.1 |

## B    CLUSTERING ANALYSIS ON CELEBA

We present, in Figure 6, the matching accuracy between the ground-truth environments and the environments discovered with our method on the validation set of CelebA for different layers of the VGG-19. As on Waterbirds, we notice that the best result is obtained when using the layer *conv5_1*.

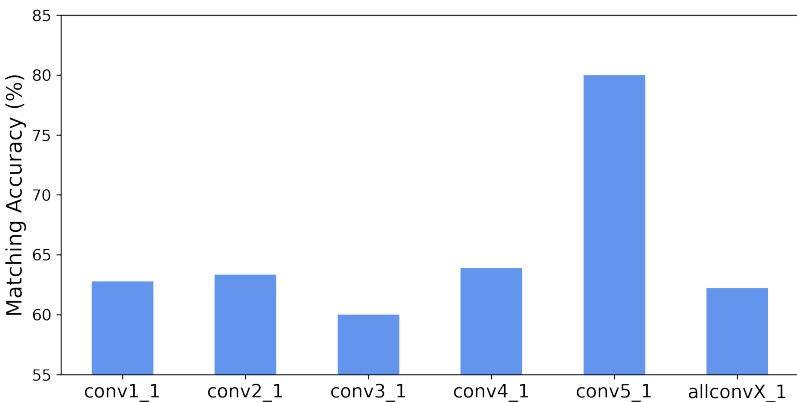

Figure 6: **Impact of the layer choice to extract features on CelebA**. We show the matching accuracy between the ground-truth environments on the validation set CelebA and the discovered ones with GRAMCLUST when using different VGG-19 layers. The result denoted *allconvX_1* is obtained when using all the layers *conv1_1, conv2_1, conv3_1, conv4_1, conv5_1* in our method.

