# OpenReview forum: "Take One Gram of Neural Features, Get Enhanced Group Robustness"
_ICLR.cc/2023/Conference — Submitted to ICLR 2023_

### Official Review · Reviewer_Nwqa · 2022-10-24

**Confidence:** 3
**Correctness:** 2
**Technical Novelty And Significance:** 1
**Empirical Novelty And Significance:** 1
**Recommendation:** 3

**Clarity, Quality, Novelty And Reproducibility:**

Clarity: could be greatly improved (see above)
Novelty: not novel
Reproducibility: The authors have provided the code but I did not go through the code. Based on the paper, there are questions around the baselines and HP tuning.

**Strength And Weaknesses:**

Strength:
- This is an important problem.
- The proposed method seems to be mostly competitive with the other baselines
- Their proposed method does not need group labels for the validation set (although the paper does not provide any clear answer as to why this is the case)
Weaknesses:
- The proposed method is computationally much more expensive than the baselines. I strongly recommend the authors to give a more comprehensive comparison between their methods and the baselines with respect to the computation burden.
- The fact that we know unsupervised clustering and two step training procedures with GDRO could result in better worst case performance has already been observed. As a result the overall results of the paper are really incremental. Unfortunately, the paper does not shed much light on why their method is superior or what are the limitations of their approaches? For example, at one point the authors mention that gram matrices seem to encode "textures and color palette" rather than “structure” in the image. So, does this mean that the proposed method works for cases where group labels or spurious correlations relate to these? I think these are more important questions to answer rather than just showing that the method works better than very similar previously proposed methods on some datasets.
- The proposed method, and specifically the hp tuning for it (for the first and second step) are not very well elaborated. This makes it very difficult to judge if their claims about fair comparisons are correct.
- Why is it that the method does not require labeled validation? Could it be true that other methods could also be applied without labeled validation? If not, what is so different about the approach that enables that? In fact, on all datasets GRAMCLUST-cv is better than GRAMCLUST-orig which is a bit counter-intuitive. I suspect this might be due to some counter-intuitive hp tuning. Note that the explanations around the mismatch in the labels might be true for water-birds, but is not very applicable to CelebA or Coco dataset.
- What is so special about VGG-19 and why should it be sued for Gram matrix? Would the result still hold if one uses other network architectures, e.g. resnet? other pre-trainings? This is so mysterious that all results are only presented with VGG-19 without any intuition/understanding.
- The JTT numbers are much lower than what they should be (reported in other papers). I suspect it is due to footnote 3. But I am not sure why footnote 3 is applicable and fair. With those numbers I do not believe that gram matrix approaches can show much gain over the baselines. I also noted that the gram matrix performances have relatively larger std compared to the baselines, which might mean that the methods are not very stable and their performance depends on the random seeds.
- Compared to meanvar, the proposed gram matrix solution is very computationally intensive with not significant gains over waterbirds and coco on p.
- The definition of matching accuracy is not provided (at least for the sake of completeness). Also, could you provide more details on the correlation between matching accuracy and the final performance?
- For figure 4 only the results for water-birds are provided while in that case the ground truth E=2 (which is the smallest possible number fo clusters). Could you please provide evidence on cases where the ground truth E is different than the minimum, e.g. Coco on P data, to better illustrate the trade-off?

**Summary Of The Paper:**

The paper proposes a new method for improving the robustness of learning neural networks with respect to group correlations using the Gram matrix that is extracted from the output of different layers in a learned neural network.

**Summary Of The Review:**

The paper's contributions are marginal without adding much to the fundamental understanding of why the method works better than other methods (this part is questionable based on the lower than expected numbers on JTT), why it does not need validation group labels and what are the limitations of it (e.g. are there any spurious correlations that this method cannot capture)?

---

> ### Author Response · Authors · 2022-11-17
> **Response to Reviewer Nwqa (1/2)**
>
> We thank you for your comments. We address them in detail below.
>
> &nbsp;
> > **The proposed method is computationally much more expensive than the baselines.**
>
> In the partition stage, computing Gram matrices from neural features can indeed be more expensive than some other baselines such as JTT which partition a dataset into correct and erroneous predictions. Nevertheless, the computational increase only happens once before training the robust classifier. Additionally, we proposed at the end of section 3.2 to apply random projection to reduce dimensions that drastically helps to scale to larger datasets. Our experiments were run on a single 2080ti GPU. Finally, note that in the second stage of training the robust classifier, our method is equivalent to any other methods training with GroupDRO loss.
>
> &nbsp;
> > **The fact that we know unsupervised clustering and two step training procedures with GDRO could result in better worst case performance has already been observed. As a result the overall results of the paper are really incremental.**
>
> Our contributions do not lie in demonstrating that two-step training procedures with GDRO improve worst-group accuracy. We show instead that clustering a training dataset based on Gram matrices provides state-of-the-art performances in group robustness for image classification. This claim is supported by results shown in Table 1 where we compare to other methods assuming no access to group labels in training. In addition, our second contribution is to select the best hyperparameters based on a validation set augmented with our inferred pseudo-groups. To our knowledge, this is the first approach (GramClust-cv) in group robustness that alleviates the need of group labels on the validation set.
>
> &nbsp;
> > **Unfortunately, the paper does not shed much light on why their method is superior or what are the limitations of their approaches?**
>
> In Section 4.3 and Table 2, results demonstrate the superiority of Gram matrices in addressing group robustness over using penultimate features ('Standard’) or simple statistics ('MeanVar’). The improvement is highly significant on the most complex dataset CelebA. This is because Gram matrice provide more information
> than MeanVar as their diagonals already contain the information about the channel-wise mean and variance of the deep features (page 8). As detailed in Section 3.2 and by the reviewer here, there is a computational increase due to the storing of Gram matrices, but we mitigate this limitation by applying random projections to reduce dimensions.
>
> &nbsp;
> > **For example, at one point the authors mention that gram matrices seem to encode textures and color palette rather than structure in the image. So, does this mean that the proposed method works for cases where group labels or spurious correlations relate to these?**
>
> * Gram matrices are first and foremost a second-order correlation matrix of neural features, before its successful application in Neural Style Transfer. As many identified spurious correlations in computer vision are related to background, texture or physical attributes (see refs, page 1), our approach addresses group robustness in these cases by clustering images based on their Gram matrix features. As pointed out in the introduction, matching Gram matrices is shown to be equivalent to aligning distribution by minimizing MMD distance. This equivalence reinforces their use to discover visual-based environments.
> * To better study the link between the use of Gram matrices and dataset biases, we evaluate within Section 4.4 and Fig. 3 the matching accuracy between our predicted environments and true environments on the Waterbirds dataset. For our original method (conv5_1), we are able to recover almost perfectly the true groups (93.41% matching accuracy).
>
> &nbsp;
> > **Why is it that the method does not require labeled validation? Could it be true that other methods could also be applied without labeled validation? If not, what is so different about the approach that enables that?**
>
> In our experiments, we observed that the Gram matrix-based clustering was effective enough to alleviate the need of ground-truth environments labels in the validation set to select good hyperparameters. Instead, we leverage pseudo-group labels inferred by our approach on the validation set. This proposed hyperparameter tuning method could indeed be applied to other methods. Nevertheless, our results in Table 1 show that the variation GramClust-orig, which uses true validation labels for model selection,, already improved other approaches on worst-group accuracy.

---

> > ### Author Response · Authors · 2022-11-17
> > **Response to Reviewer Nwqa (2/2)**
> >
> > > **Note that the explanations around the mismatch in the labels might be true for water-birds, but is not very applicable to CelebA or Coco dataset.**
> >
> > The Section 4.5 elaborates on the improved results of GramClust-orig over GroupDRO, which may be counterintuitive at first as GroupDRO used the true group labels. Hence, the explanation in this section does not aim at explaining the difference in performances between GramClust-orig and GramClust-cv, but rather between GramClust-orig over GroupDRO.
> >
> > &nbsp;
> > > **What is so special about VGG-19 and why should it be sued for Gram matrix? Would the result still hold if one uses other network architectures, e.g. resnet? other pre-trainings?**
> >
> > Although dating from 2015, VGG is still frequently used for image style transfer (see end of page 6): out of 8 papers on this topic at CVPR’22, 6 rely on VGG, 1 on ResNet and 1 on a ViT. Our preliminary experiments on other network architectures for the identification model revealed degraded final performance, such as below with a ResNet-50:
> >
> > | Architecture + Layer | RN50-conv1_1 | RN50-conv2_1 | RN50-conv3_1 | RN50-conv4_1 | VGG19-conv5_1 |
> > |----------------------|:------------:|:------------:|:------------:|:------------:|:-------------:|
> > | Worst-group Accuracy |     78.3%    |     71.8%    |     83.2%    |     84.1%    |   **85.3%**   |
> >
> > &nbsp;
> > > **The JTT numbers are much lower than what they should be (reported in other papers). I suspect it is due to footnote 3. But I am not sure why footnote 3 is applicable and fair. With those numbers I do not believe that gram matrix approaches can show much gain over the baselines.**
> >
> > When using JTT baseline from author’s code, we observed that the reported scores were associated with models trained with early stopping, in fact after only 3 epochs of training. While this early stopping helped to obtain better worst-group accuracy, it also degrades the average accuracy performance. For instance, on Waterbirds, Table 1 in the original paper [Liu et al., 2021] reports 93.3% average accuracy, compared to 97.3% to ERM baseline, both in the original paper and in our submission. The drop in accuracy is significant: -4.0 points. We argue that it is not a suitable property for a model to have such a drop in average predictive performance and prefer models with high accuracy both in average and on the worst group of the test set (page 7). Moreover, in JTT original paper, early stopping is performed by using a small validation set annotated with true group labels, while GramClust-cv does not need any group label in training.
> >
> > &nbsp;
> > > **I also noted that the gram matrix performances have relatively larger std compared to the baselines, which might mean that the methods are not very stable and their performance depends on the random seeds.**
> >
> > For each dataset, GramClust has a comparable std on average accuracy (Waterbirds, CelebA) and in-distribution accuracy (COCO-on-P) to other methods: between +-0.1 and +-0.3. Regarding worst-group accuracy, GramClust-cv indeed achieves a higher std (+-1.1) than JTT, EEIL and GroupDRO (=-0.3) but remains lower than GEORGE (+-2.0). On CelebA, GramClust-cv std (+-1.9)  is in a similar range to stds of other methods such as ERM (+-1.5) and GroupDRO (+-2.0). Finally, stds are nearly similar for all methods, including GramClust, on CelebA: between +-0.3 and +-0.4.
> >
> > &nbsp;
> > > **Compared to meanvar, the proposed gram matrix solution is very computationally intensive with not significant gains over waterbirds and coco on p.**
> >
> > In Section 4.3 and Table 2, results demonstrate the superiority of Gram matrices in addressing group robustness over using penultimate features ('Standard’) or simple statistics ('MeanVar’). The improvement is highly significant on the most complex dataset CelebA (+8.1 point) and also non-negligible on COCO-on-Places (+1 point). This is because the Gram matrix provides more information than MeanVar as its diagonal already contains the channel-wise mean and variance of the deep features (page 8).
> >
> > &nbsp;
> > > **The definition of matching accuracy is not provided (at least for the sake of completeness). Also, could you provide more details on the correlation between matching accuracy and the final performance?**
> >
> > The matching accuracy is the percentage of samples whose predicted group corresponds to its true group. We thank you for pointing out this oversight, we have added it in the revised version. In our experiments, we observe a strong correlation between matching accuracy and worst-group accuracy for CelebA and COCO-on-Places. For Waterbirds, the correlation holds until a certain point, due to the presence of confusing samples showcased in Figure 5.

---

### Official Review · Reviewer_4Q7E · 2022-10-25

**Confidence:** 3
**Correctness:** 4
**Technical Novelty And Significance:** 2
**Empirical Novelty And Significance:** 3
**Recommendation:** 6

**Clarity, Quality, Novelty And Reproducibility:**

I think the high level idea is intuitive, but limited on novelty. The experimental results are strong and make for a good contribution. The paper is written well.

**Strength And Weaknesses:**

Pros:
- The idea of clustering to identify group information is a natural one to try out.
- The paper shows that one can get competitive performance on common benchmarks even with no knowledge of the underlying groups, even against baselines that do make use of group label info in the validation set. This I think is a significant empirical contribution.

Cons:
- Unfortunately, the idea lacks sufficient formalism. While it is intuitive that clustering the data based on learned embeddings could help discover subgroups, I think the paper would be more compelling if the authors are able to provide formal framework for characterizing spurious correlations and identifying scenarios where applying a simple k-means clustering to the training data would uncover the right subgroups.
- Determining the right number of clusters remains an important challenge. The authors show robustness to the number of clusters on one of the datasets (do these trends hold for the others too?), but in general, I worry that mis-specifying the number of groups could adversely impact the subsequent DRO step. Since DRO optimizes for worst-case accuracy, doing so with a larger number of groups than needed may adversely impact average accuracy. Could you show a similar plot as Fig 4 for average accuracy too?

Other questions:
- In the experiments, some times GramClust-cv (no group labels) outperforms GramClust-orig (groups labels in val set). This happens not only in Waterbirds (where the authors point to possible noise in group annotations), but also in CelebA. Is there a reason you've identified for this?
-  In your implementation of Group DRO from Sagawa et al., did you implement the per-group regularized updates in eq (5) in their paper (which I think is what they use in their experiments), or the online version in Algorithm 1 in their paper?

Additional related literature: Kirichenko et al. Last Layer Re-Training is Sufficient for Robustness to Spurious Correlations. ArXiv:2204.02937.



**Summary Of The Paper:**

One of the challenges in training over-parameterized neural networks is that they may overfit to "spurious features", resulting in poor generalization performance on minority subgroups. A popular approach to alleviate such biases is the use of group distributionally robust optimization (DRO). However, prior DRO approaches required group labels to be available either in the training or validation set. This paper proposes a method to automatically discover the underlying subgroups, targeting practical use cases where the underlying spurious correlations are unknown.

The proposed method first trains an embedding model, and clusters the data points into groups based by computing (dot product) similarity between embeddings. Group DRO is then applied treating the identified clusters as subgroups. The proposed method uses no group label information, yet is competitive with methods that use group labelings in a validation set.



**Summary Of The Review:**

Overall, intuitive idea, strong experiments, but limited novelty. The paper would benefit from some minimal theoretical justification.

---

> ### Author Response · Authors · 2022-11-17
> **Response to Reviewer 4Q7E**
>
> We thank you for your detailed comments as well as your additional literature references that have been added to the revised version of our paper. Our answers to your questions are provided below.
>
> &nbsp;
> ### Formalism
> Li *et al.* (2017) demonstrate that matching Gram matrix neural features is actually equivalent to aligning distributions by minimizing MMD (maximum mean discrepancy distance) with a polynomial kernel between distributions. Our K-means approach could therefore be written in terms of MMDs, i.e. discovering clusters of similar feature distributions that are likely candidates for environments.
>
> &nbsp;
> ### Number of clusters and DRO
> The GroupDRO optimization is done following the online algorithm from the original paper (Algorithm 1). At each iteration, data points are sampled randomly for each group. The update w.r.t. the previous data points is proportional to the mass of the group. The mass is computed in such a way that it is large for large loss group values. Therefore, all the groups participate in the updates of the model’s parameters mitigating the pathological case with a cluster containing a few hard data points associated with high losses.

---

### Official Review · Reviewer_9Mcs · 2022-10-26

**Confidence:** 4
**Correctness:** 4
**Technical Novelty And Significance:** 3
**Empirical Novelty And Significance:** 3
**Recommendation:** 6

**Clarity, Quality, Novelty And Reproducibility:**

The paper is overall well written and clear. The idea of using Gram matrices to do the clustering in the first state is novel and also leads to improved performance.

**Strength And Weaknesses:**

The idea of using Gram matrices which are supposed to capture the style and texture features of the images and hence, can capture the spurious features is a simple and interesting idea.

They propose to use the groups learned for even the validation set but I wanted to check with the authors that even with other previous methods, we can do the same thing?

Also, is there any guarantee that we will indeed learn the natural groups of the data. Are there any examples of the dataset where the groups contain divisions based on non-style and texture based features and hence, won’t be captured well by the Gram matrices based clustering?

The authors also compare their method to using last layer features for clustering. In their identifier model, they have also trained the model for only a few iterations so that the model only capture easy features. Did the authors use the same early stopped model for the last layer features while doing this experiment? It would be good to separate out the gains coming from early stopping vs. using Gram matrices for clustering.

It is quite interesting to see that increasing the number of clusters does not hurt their performance. I would imagine that increasing the number of clusters can maybe form some clusters which only contain hard data points and having low loss on them can hurt the overall performance. Do the authors have some intuition on what kind of clusters does this method find when the set the number of clusters to be large and why it does not hurt the performance?

**Summary Of The Paper:**

Recently, it has been observed that although neural networks achieve very good accuracy on average over the dataset but they can have very low accuracy on certain subgroups of the distribution. There have been many works on trying to fix this gap. This paper proposes an algorithm to improve the accuracy on different subgroups of the distribution and show improved performance on many benchmark datasets as compared to many existing baseline methods. The benefit to their approach is they do not require information about the group labels and learn them using unsupervised clustering. The unsupervised clustering has been used previously but they propose to use clustering over the Gram matrices of the activations of the intermediate layers which are shown to capture the texture and style patterns of images and hence can capture the groups divisions well. They use these clusters to perform group distributionally robust optimization in the second stage. Moreover, they propose to use the learned clusters of the validation set for tuning the hyper parameters as well and hence, do not even require knowledge of these subgroups over the validation set.

**Summary Of The Review:**

I like the overall idea of the paper. This work also improves performance on many benchmark datasets. The only concern is whether the gain is coming from using Gram matrices or the early stopped model.

---

> ### Author Response · Authors · 2022-11-17
> **Response to Reviewer 9Mcs**
>
> We thank you for your detailed comments. Our answers to your questions are provided below.
>
> &nbsp;
> ### Hyperparameter selection and other methods
> Our proposed cross-validation with pseudo-group to perform the hyperparameter selection could indeed be applied to other methods. Nonetheless, GramClust-orig already obtains the highest worst-group accuracy on all datasets with a significant margin (more than 20 points on CelebA), with GramClust-orig using true groups annotation for hyperparameter selection on the validation set like other methods.
>
> &nbsp;
> ### Gram matrix and true group annotation recovery
> Datasets in the group robustness literature are mainly vision datasets with spurious correlations based on various visual features (hair color, backgrounds, set of backgrounds).
>
> In Figure 3 (in the main paper) and Figure 6 (in the appendix), the Hungarian matching score of our method (which uses conv5_1 layer) shows that we are able to recover most of the true-group annotation in the case of background correlation (Waterbirds) and physical attribute (CelebA).
>
> &nbsp;
> ### Training the identification model
> We conducted experiments with the identification model trained during the full 300 epochs and it impacted the final performance. We provide preliminary results on Waterbirds by training a ImageNet-pretrained ResNet-50 fine-tuned during 1 epoch vs. 300 epochs.
>
> | Training   | Worst-group Acc |
> |------------|-----------------|
> | 1 epoch    |           76.2% |
> | 300 epochs |           72.3% |
>
> Training during only a few epochs enables the model to capture high-level features. By overfitting the identification model, features become progressively object-focused, which degrades the quality of the predicted environments using our approach.
>
> &nbsp;
> ### Number of clusters and performance
> The implementation of the GroupDRO optimization problem is done following the online optimization algorithm in the original paper (Algorithm 1). At each iteration, data points are sampled randomly for each group. The update w.r.t. the previous data points is proportional to the mass of the group. The mass is computed in such a way that it is large for large loss group values. Therefore, all the groups participate in the updates of the model’s parameters mitigating the pathological case with a cluster containing a few hard data points.

---

### Official Review · Reviewer_nkU5 · 2022-10-27

**Confidence:** 3
**Correctness:** 2
**Technical Novelty And Significance:** 2
**Empirical Novelty And Significance:** 2
**Recommendation:** 5

**Clarity, Quality, Novelty And Reproducibility:**

The clarity is good. Some examples are adopted to clarify the problem clearly.

The quality needs to be better. Some descriptions are not rigorous.
For example, the author mentioned that ``This results show empirically that our proposed approach, using Gram matrices of feature to discover pseudo-groups, which are then used for robust optimization and hyperparameter cross-validation, is very effective for group robustness. It also supports that Gram matrices are well suited to capture various types of dataset biases (background for Waterbirds, physical attribute in CelebA, multiple backgrounds in COCO-on-Places-224). "
The Gram matrices are supposed to extract texture features. The higher accuracy can't prove the Gram matrices are well suited to capture various types of dataset biases (background for Waterbirds, physical attribute in CelebA, multiple backgrounds in COCO-on-Places-224). The connection between the function of Gram matrices between the background, physical attribute, and other biases should be verified with other experiments.

The novelty is weak.
The reproducibility is good. The author provides a detailed experiment setting in the manuscript.


**Strength And Weaknesses:**

Strength:
The proposed method can achieve best worst-group (w-g) test accuracy.


Weaknesses:
The novelty is weak. The proposed method is just a combination of gram matrices and group robustness optimization essentially.
The proposed method can't achieve the best performance on average accuracy.
The proposed method is not validated on the large-scale dataset.

**Summary Of The Paper:**

In this paper, the author combines the existing gram matrices into the group robustness classification. The whole pipeline can be summarized into two steps: the author first adopts the clustering techniques to cluster the dataset samples into several groups with the gram matrices features; then,  cluster groups with the pseudo-group labels are integrated into the existing group robustness optimization framework.
The proposed method is validated on three datasets. The proposed method only can achieve higher test accuracy on worst-group (w-g) but not achieve the best performance on average (avg) test accuracy.

**Summary Of The Review:**


Overall, novelty is weak. The experiment results only can support part of the claims or the proposed method.

---

> ### Author Response · Authors · 2022-11-17
> **Response to Reviewer nkU5**
>
> We thank you for your detailed feedback, and address your concerns below.
>
> &nbsp;
> ### Novelty
>
> In computer vision, Gram matrices have been recently used to transfer the style of an image to another existing image and in out-of-distribution detection (page 3). Our main contribution is to show that they can also be applied to group robustness where clustering images based on Gram matrix neural features leads to state-of-the-art results on multiple datasets. In Table 1, comparative results between our proposed approach and methods with group labels (IRM, GroupDRO) show that using Gram matrices alleviates the need of ground-truth group labels altogether to obtain strong performance on worst-group accuracy while maintaining high average accuracy. We also design a novel hyper-parameter selection method that consists in leveraging pseudo-group labels inferred by our approach on the validation set.
>
> &nbsp;
> ### Worst-group and average accuracy
> In group robustness, the goal is to ensure robust predictive performances across groups within a dataset. It is measured by the worst-group accuracy. Previous works, e.g., GroupDRO, JTT, GEORGE, already observed that methods in the group robustness literature improving worst-group accuracy also suffer from a slight decrease in average accuracy compared to ERM-trained models. Nevertheless, in Table 1 we observe that the average accuracy of GramClust is on-par with other group robustness methods also assuming no group labels on Waterbirds and CelebA, and even outperform them on COCO-on-Places.
>
> &nbsp;
> ### Datasets
> We validated our approach on 3 datasets standardly used in group robustness literature (see GroupDRO, JTT, GEORGE). Among them, CelebA is composed of more than 200k images and constitutes a dataset with (1) the larger amount of images/samples used in the literature and (2) a naturally occurring spurious correlation between hairstyle and gender. We agree that providing another large-scale dataset could be an interesting asset for the community as a future work, but it would be a work on its own.
>
> &nbsp;
> ### Gram matrices and dataset biases
> To better study the link between the use of Gram matrices and dataset biases such as background and physical attributes, we evaluate in Section 4.4 and Fig. 3 the matching accuracy between our predicted environments and true environments on the Waterbirds dataset. For our original method (conv5_1), we are able to recover almost perfectly the true groups (93.41% matching accuracy). The improved worst-group accuracy is also an indirect evidence that environments have been correctly discovered with Gram matrices, as GramClust’s performance is close to GroupDRO’s performance, the latter using the true environments.

---

### Author Response · Authors · 2022-11-17
**General Response and Update**

We would like to thank the reviewers for their careful reading of our work and their detailed comments. Incorporating their feedback, we have uploaded a revised manuscript including new references, the definition of matching accuracy and fixed typos. We hope our responses and revisions address all reviewers’ concerns, and we would greatly appreciate any further comments and clarifications that we can make.

---

### Decision · Program_Chairs · 2023-01-20

**Decision:**

Reject

**Justification For Why Not Higher Score:**

A higher score could have been given if more compelling explanations of the most interesting experimental results were given,  such as better theoretical support for the method and a clearer explanation of behavior wrt number of clusters/validation on clusters.

**Justification For Why Not Lower Score:**

N/A

**Metareview: Summary, Strengths And Weaknesses:**

There is agreement between the reviewers that the paper is clear and the experimental results show that the method is competitive with respect to the baselines.

Nonetheless, there is also agreement on the main weaknesses of the paper:
1- at a high level, the paper proposes to apply a well-known clustering method on top of an already published idea that two-step clustering/groupDRO could allow for robust learning in the absence of group labels; the overall novelty is therefore low, from both the conceptual and technical points of views. The relationship between aligning Gram matrices and MMD with polynomial kernel is here but it is still not an end-to-end argument regarding its performance in the authors' setting.
2- while the experimental results are promising, there is still a lack of explanation as to why it is so. The evolution of the results wrt number of clusters, or why cross-validation with clusters rather than group labels are very open questions and yet are important experimental results. The author response did not bring convincing additional arguments.